Plant–insect interactions from Middle Triassic (late Ladinian) of Monte Agnello (Dolomites, N-Italy)—initial pattern and response to abiotic environmental perturbations

Wappler Torsten 1 twappler@uni-bonn.de
Kustatscher Evelyn 2 3
Dellantonio Elio 4
1 Steinmann Institute, University of Bonn , Bonn , Germany
2 Naturmuseum Südtirol , Bozen/Bolzano , Italy
3 Department für Geo- und Umweltwissenschaften, Paläontologie und Geobiologie, Ludwig-Maximilians-Universität and Bayerische Staatssammlung für Paläontologie und Geobiologie , München , Germany
4 Museo Geologico delle Dolomiti , Predazzo (TN) , Italy
DiMichele William
Electronic publication date: 2015 Apr 28
Publication date: 2015
Volume: 3
Electronic Location ID: e921
Received 2015 Feb 7; Accepted 2015 Apr 8
Copyright: © 2015 Wappler et al.
Copyright year: 2015
Copyright holder: Wappler et al.
License: This is an open access article distributed under the terms of the Creative Commons Attribution License, which permits unrestricted use, distribution, reproduction and adaptation in any medium and for any purpose provided that it is properly attributed. For attribution, the original author(s), title, publication source (PeerJ) and either DOI or URL of the article must be cited.
License URL: https://creativecommons.org/licenses/by/4.0/

Keywords: Plant–animal interactions, Plant fossils, Italy, Longobardian, Southern Alps, Volcanic activity

Funding: Comune di Predazzo (Provincia di Trento) Promotion of Educational Policies, University and Research Department of the Autonomous Province of Bolzano—South Tyrol German Science Foundation 1496/8-1 The work has been supported by the Comune di Predazzo (Provincia di Trento). This study is part of the project “The Permian-Triassic ecological crisis in the Dolomites: extinction and recovery dynamics in Terrestrial Ecosystems” financed by the Promotion of Educational Policies, University and Research Department of the Autonomous Province of Bolzano—South Tyrol. TW is supported by a Heisenberg grant from the German Science Foundation (1496/8-1). The funders had no role in study design, data collection and analysis, decision to publish, or preparation of the manuscript.

==============================
The Paleozoic–Mesozoic transition is characterized by the most massive extinction of the Phanerozoic. Nevertheless, an impressive adaptive radiation of herbivorous insects occurred on gymnosperm-dominated floras not earlier than during the Middle to Late Triassic, penecontemporaneous with similar events worldwide, all which exhibit parallel expansions of generalized and mostly specialized insect herbivory on plants, expressed as insect damage on a various plant organs and tissues. The flora from Monte Agnello is distinctive, due to its preservation in subaerially deposited pyroclastic layers with exceptionally preserved details. Thus, the para-autochthonous assemblage provides insights into environmental disturbances, caused by volcanic activity, and how they profoundly affected the structure and composition of herbivory patterns. These diverse Middle Triassic biota supply extensive evidence for insect herbivore colonization, resulting in specific and complex herbivory patterns involving the frequency and diversity of 20 distinctive damage types (DTs). These DT patterns show that external foliage feeders, piercer-and-suckers, leaf miners, gallers, and oviposition culprits were intricately using almost all tissue types from the dominant host plants of voltzialean conifers (e.g., Voltzia), horsetails, ferns (e.g., Neuropteridium, Phlebopteris, Cladophlebis and Thaumatopteris), seed ferns (e.g., Scytophyllum), and cycadophytes (e.g., Bjuvia and Nilssonia).

Introduction

Continental arthropods and vascular plants have been major elements of terrestrial ecosystems worldwide for nearly 400 million years, and their varied ectophytic and endophytic associations can provide a unique and direct record of the plant–insect interactions in the past (e.g., Labandeira & Currano, 2013). In 2006, Labandeira proposed four pulses of herbivore expansion, where the observed Palaeozoic arthropod herbivory patterns—covering the first two phases—are mainly expressed by damage patterns caused by mites and apterygote and/or basal pterygote herbivores on pteridophyte and basal gymnospermous plant hosts and are profoundly different from those that originated after the end-Permian mass extinction (Labandeira, 2006a; Labandeira, 2006b). Preliminary work on plant–insect interactions from early to late Permian floras of the Southern Alps indicates a moderately diverse pattern of damage occurring in a variety of habitats prior to the P-Tr crisis (T Wappler, pers. obs., 2013), penecontemporaneous with similar events in US Southwest (e.g., Schachat et al., 2014), Gondwana (Adami-Rodrigues et al., 2004; Adami-Rodrigues, Iannuzzi & Pinto, 2004; Cariglino & Gutiérrez, 2011; Gallego, Cúneo & Escapa, 2014; Iannuzzi & Labandeira, 2008; Prevec et al., 2009; Slater, McLoughlin & Hilton, 2012), or Cathaysia (e.g., Glasspool et al., 2003). Nevertheless, herbivory expansion 2 was profoundly disrupted by environmental perturbations at the P-Tr boundary. The Early Triassic has been traditionally viewed as an unusual time marked by suppressed origination rates and low diversity (e.g., Benton & Emerson, 2007) generally attributed to the effects of extreme environmental conditions inflicted on Early Triassic ecosystems (e.g., Looy et al., 1999; Grauvogel-Stamm & Ash, 2005; Roopnarine et al., 2007; Tong et al., 2007) but taphonomical biases cannot be excluded at least for European floras (Kustatscher et al., 2014). In general, records of Early Triassic insects or of insect damage on plants are scant worldwide (comp. Table 1), so little is known about the mechanics and timing of diversification of this ecologically important group following the end-Permian mass-extinction event (Kustatscher et al., 2014; Labandeira & Currano, 2013). Shcherbakov (2008a) even concluded that the entire class of insects was strongly reduced in diversity at the P-Tr boundary but following the end-Permian biotic crisis insect faunas already contained many elements common to modern insects (e.g., Aristov et al., 2013; Béthoux, Papier & Nel, 2005; Shcherbakov, 2008b; Lukashevich et al., 2010; Żyła et al., 2013; Haig et al., 2015) building the nucleus for the onset of the third pulse of herbivore expansion, coupled with an impressive adaptive radiation of herbivorous insects. Their associations with plants became significantly diverse being major elements for keystone communities in terrestrial ecosystems worldwide (e.g., Ash, 2014; Grauvogel-Stamm & Kelber, 1996; Kustatscher et al., 2014; Labandeira, 2006a; Labandeira, 2006b; Labandeira & Currano, 2013; McLoughlin, 2011; Moisan et al., 2012; Pott et al., 2008; Scott, Anderson & Anderson, 2004). Simultaneously, a major three-phased floral change has been proposed for Europe and probably worldwide (e.g., Grauvogel-Stamm & Ash, 2005). The first stage lasted from the Induan to early Anisian, which in Europe is characterized by a “survival” interval dominated by the lycopsid Pleuromeia Corda ex Giebel (1853) and conifers coupled with relatively low levels of plant–insect interactional diversity (Kustatscher et al., 2014); this is followed by a “recovery” interval characterized by the resurgence of lycopsids, sphenophytes, ferns, cycadophytes, conifers, ginkgophytes and seed ferns. The second stage occurred from the late Anisian to the Carnian. The third covers the Norian and Rhaetian stages, which is pivotal to understanding the evolution of trophically modern ecosystems (e.g., Benton, 2010; Labandeira, 2006b; Labandeira & Currano, 2013).

Table 1 Arthropod damage on Triassic plants.

List of published records of arthropod damage on Triassic plants.

Study	Age	Formation and locality	Damage type	
Nathorst (1876); Nathorst (1878)	Rhaetian (Late Triassic)	Pålsjö, Scania, Sweden	• Possible oviposition scars on Podozamites	
Ghosh, Kar & Chatterjee (2015)	Norian/Rhaetian (Late Triassic)	Parsora Formation (Dhaurai Hill beds); South Rewa Gondwana Basin, central India	• Disc-like galls on Dicroidium hughesii	
Walker (1938); Ash (1997); Ash (1999); Ash (2000); Ash (2001); Ash (2005); Ash & Savidge (2004); Ash (2014); Creber & Ash (2004)	Norian (Late Triassic)	Chinle Formation, Petrified Forest National Park, Arizona, USA	• Marginal and non-marginal feeding traces on Cynepteria, Marcouia, Zamites, Sphenopteris, Macrotaeniopteris, Dechellyia, Nilssoniopteris
• Possible oviposition scars and insect eggs on Dechellyia, ?Equisetites
• Coprolite-bearing borings in Itopsidema, Araucarioxylon, Schilderia	
Adami-Rodrigues, Gnaedinger & Gallego (2008)	Norian (Late Triassic)	El Tranquilo Group, Laguna Colorada Formation; Santa Cruz, Argentinia	• Specific and complex herbivory patterns of several FFG’s	
Feng et al. (2014); Hsü et al. (1974)	Keuper (Late Triassic)	District Yungjen, Yunnan, China	• Crescent-shape bite marks on Mixopteris • Intense skeletonization Dictyophyllum nathorstii	
Gallego et al. (2003); Gallego et al. (2004); Gnaedinger, Adami-Rodrigues & Gallego (2007); Gnaedinger, Adami-Rodrigues & Gallego (2008); Gnaedinger, Adami-Rodrigues & Gallego (2014)	Carnian-Norian (Late Triassic)	La Ternera Fm. (Quebrada La Cachivarita locality; La Ternera hill area, Copiapó Province), and the Las Breas Fm. (Punta del Viento locality, Vicuña, Elqui Province), Chile	• Oviposition scars on Heidiphyllum, Pseudoctenis, Taeniopteris	
Strullu-Derrien et al. (2012)	Carnian (Late Triassic)	De Geerdalen Formation; Hopen Island, Svalbard Archipelago	• Aggregations of pellets or coprolites within bennettitalean roots
• Gall-like structures within the cortical or pith tissues of the larger (probable bennettitalean) axes	
Rozefelds & Sobbe (1987); Tillyard (1922); Webb (1982)	Carnian (Late Triassic)	Blackstone Formation, Ipswich Coal Measures Group; Sydney Basin, New South Wales, Australia	• Possible oviposition scars and insect eggs on Nilssoniopteris
• Possible galls or eggs on Dictyophyllum
• Mining structures on Heidiphyllum, Ginkgoites	
Meller et al. (2011); Pott et al. (2008); B Aschauer & T Wappler, 2012, unpublished data	Carnian (Late Triassic)	Lunz Formation; Lunz am See, eastern Northern Calcerous Alps, Austria	• Possible oviposition scars and insect eggs on Nilssoniopteris
• Possible mining structures on Nilssonia
• Marginal and non-marginal feeding traces on Nilssoniopteris, and other bennettitalean leaves	
Moisan et al. (2012)	Carnian (Late Triassic)	Madygen Formation; Turkestan Mountains, southwestern Kyrgyzstan, Central Asia	• Oviposition scars on Isoetites	
Anderson & Anderson (1983); Anderson & Anderson (1985); Anderson & Anderson (2003); Labandeira & Anderson (2005); Scott, Anderson & Anderson (2004)	Carnian (Late Triassic)	Molteno Formation; Karoo Basin, KwaZulu- Natal, Eastern Cape and Northern Cape, South Africa	• Specific and complex herbivory patterns involving the frequency and diversity of 79 distinctive damage types (DTs) on about 220 whole-plant species (liverworts, lycopods, horsetails, ferns, cycads, peltasperms, corystosperms, hamshawvialeans, ginkgoaleans, voltzialean conifers, bennettitaleans, gnetophytes)	
Linck (1949); Roselt (1954)	Carnian/Ladinian (Upper/Middle Triassic)	Bedheim, Germany	• Borings in Dadoxylon
• Possible oviposition scars on Equisetites	
Geyer & Kelber (1987); Kelber & Geyer (1989)	Upper Ladinian (Middle Triassic)	Lettenkohle of Alsace, France; Lower Keuper of Franconia, Germany	• Crescent-shape bite marks on Schizoneura, Taeniopteris
• Possible oviposition scars and insect eggs on Equisetites	
Heer (1877)	Ladinian (late Middle Triassic)	Neuewelt, Lettenkohle, Switzerland	• Possible oviposition scars on Equisetites	
Minello (1994)	Ladinian (Middle Triassic)	Xinigua, Rio Grande do Sul, Santa Maria Formation (Rosario do Sul Group), Brazil	• Coprolite-bearing borings in Araucarioxylon	
Grauvogel-Stamm & Kelber (1996)	Early Anisian (Early Middle Triassic)	Grès à Voltzia Formation; Grès-à-Voltzia, northern Vosges Mountains, France	• Crescent-shape bite marks on Neuropteridium
• possible eggs entangled in plant debris	
McLoughlin (2011)	Anisian—Ladinian (Middle Triassic)	Wivenhoe Hill, Esk Trough, Esk Formation; Queensland, Australia	• Oviposition scars on Taeniopteris	
McLoughlin (2011)	Olenekian—Anisian (late Early to early Middle Triassic)	Turrimetta Head, Sydney Basin; New South Wales, Australia	• Gall on Dicroidium	
Kustatscher et al. (2014)	Olenekian (Lower Triassic)	Solling Formation; Bremke and Fürstenberg, Germany	• Specific herbivory patterns involving the frequency and diversity of 8 distinctive damage types (DTs)
• External feeding damage on Tongchuanophyllum. Neuropteridium, Pelourdea
• Mid-vein gall on Tongchuanophyllum
• Linear series of lenticular or ovoidal oviposition scars on Tongchuanophyllum	

Thus, the late Middle Triassic (Ladinian) floras of the Dolomite Region in the Southern Alps of northeastern Italy provide an intriguing window into the early evidence for Herbivore Expansion 3. Ladinian floras from the Dolomites have been extensively studied in recent years (e.g., Kustatscher, Dellantonio & Van Konijnenburg-van Cittert, 2014; Kustatscher & Van Konijnenburg-van Cittert, 2005 and references therein), evidencing a dominance of conifers (Voltzia, Pelourdea), while cycadophytes, seed ferns, ferns, horsetails, and lycopsids are much rarer. Nevertheless, the flora from Monte Agnello is markedly distinct from other Ladinian floras of the Dolomites by its higher diversity and abundance in cycadophytes, seed ferns and ferns. It is currently the best documented and most diverse late Middle Triassic biota in the Alps documenting a rich vascular plant record, including moderate levels of external foliage feeding, piercing-and-sucking, galling, and ovipositional damage.

Of particular importance, from a taphonomic viewpoint, the Dolomites were subject to significant volcanic activity, beginning in the late Ladinian. Consequently, conditions for exceptional preservation were high. Although most of the volcanic complexes were submarine, locally, such as in the area of Predazzo, subaerial eruptive centers existed (Hoernes, 1912; Leonardi, 1967), which alter the natural environment to variable extents and initiate very different effects on community composition, structure, function, and successional turnover on local and regional scales (e.g., Walker & Wardle, 2014). This makes the Monte Agnello ideal for examining the response that such environmental perturbation had on community structures and offers the possibility to study the ecological expansion of interactional diversity recorded from the varied habitats.

Geological and paleontological setting

Monte Agnello (Fig. 1) represents an area that was marginally influenced by the Ladinian volcanic activity of the Predazzo volcano and is characterized by a well-preserved stratigraphic succession (e.g., Kustatscher, Dellantonio & Van Konijnenburg-van Cittert, 2014). The 250 m thick volcanic succession is composed of “explosion breccia” at the base, followed by lava breccia, and alternations of lava flows and tuffs (Calanchi, Lucchini & Rossi, 1977; Calanchi, Lucchini & Rossi, 1978; Lucchini, Rossi & Simboli, 1982). The “explosion breccia” comprises lithic fragments (calcareous, volcanic and metamorphic fragments, clastic rocks, isolated crystals), related to the Permo-Triassic volcano-sedimentary succession and the metamorphic basement (Vardabasso, 1930). The lithic fragments of the breccia are bound by carbonate and/or chlorite-serpentine cement (Calanchi, Lucchini & Rossi, 1977). The thickness of this “explosion breccia” varies between 25 m at Monte Agnello and 10 m at Censi. The volcanic succession accumulated mostly in a subaerial environment, and is related to explosive phreatic activity (e.g., bomb sags, antidunes, accretionary lapilli; Calanchi, Lucchini & Rossi, 1977; Lucchini, Rossi & Simboli, 1982).

Figure 1 Simplified geological map of the Monte Agnello area (Dolomites, N-Italy), modified from Vardabasso (1930). MA1–MA8, fossil sites.

The flora is preserved in the tuff lenses at the base of the “explosion breccia” of the volcanic succession at Predazzo (Kustatscher, Dellantonio & Van Konijnenburg-van Cittert, 2014), which hinders an appropriate stratigraphic correlation between the diffrerent sites. Considering that they are related to one or perhaps a few phreatomagmatic events within the restricted time frame of late Ladinian volcanism the possible time difference between the single localities is however very reduced. The flora is composed of a large number of fronds, stems and reproductive organs of sphenophytes, ferns, seed ferns, cycadophytes and conifers. Due to the preservation in tuff layers, the organic material is missing and sometimes the remains are preserved only as impressions. Several stem fragments belong to the sphenophytes. The ferns are represented by Osmundaceae (Neuropteridium elegans (Brongniart) Schimper in Schimper & Schenk, 1879), Matoniaceae (Phlebopteris fiemmensis Kustatscher, Dellantonio & Van Konijnenburg-van Cittert, 2014) and Dipteridaceae (Thaumatopteris sp.). For the latter two families, it is the oldest fossil occurrence to date for the Northern Hemisphere. Additional ferns of unknown botanical affinity are Cladophlebis ladinica Kustatscher, Dellantonio & Van Konijnenburg-van Cittert, 2014, Cladophlebis sp. (Osmundaceae and/or the Dicksoniaceae) and Chiropteris monteagnellii Kustatscher, Dellantonio & Van Konijnenburg-van Cittert, 2014 (Dipteridaceae?). The seed ferns are represented by leaf fragments of Scytophyllum bergeri Bornemann, 1856. The cycadophyte leaf fragments probably belong to the genera Bjuvia Florin, 1933, Taeniopteris Brongniart, 1828 and/or Macrotaeniopteris Schimper, 1869 as well as Nilssonia Brongniart, 1828 and Apoldia Wesley, 1958. The conifers are represented by shoots of Voltzia Brongniart, 1828 and Pelourdea Seward, 1917 leaves. These plants grew probably during a humid spell, recently proposed for the late Ladinian of the Dolomites (Preto, Kustatscher & Wignall, 2010 and references therein).

Material and Methods

Data collection

Fossil plant assemblages were quantitatively censused from multiple sites at the base of the “explosive breccia,” that crops out on the northwestern slope of Monte Agnello—Censi, overlying a carbonate platform of late Anisian to Ladinian age (Sciliar Dolomite). About 684 specimens have been collected from eight distinctive sites denoted by the prefixes MA 1–MA 8 (Fig. 1 and Table 2). Sample size ranges from 2 to 244 plant remains, depending primarily on the quality and accessibility of the fossils. For the quantitative study, each identifiable plant fossil was counted. Of the plant fossil specimens collected at Monte Agnello, all that were adequately preserved and exceeded a minimum size of 0.5 cm2 were examined for insect damage. Parts and counterparts were matched whenever possible to avoid duplication. When possible, all specimens were assigned to a known species or plant morphotype. All analyzed specimens are housed at the Museo Geologico delle Dolomiti, Predazzo. Specimens occurring on the same rock slab are identified by different letters following the catalogue number whereas capital letters indicate parts and counterparts of the same specimen.

Table 2 Floral and insect damage composition late Ladinian flora from Monte Agnello, Dolomites, Italy.

Species	# Leaves	% DMG	% Spec	% Gall	% Mine	% External	% PS	% Ovi	DTs	# FFGs	DTO all	DTO spec	DTO external	DT numbers	
Bjuvia cf. dolomitica	113	15.93	0.89	0.89		14.16		0.89	9	5	21	1	18	1;2;3;12;14; 17;29;80;100	
Chiropteris montagnellii	12														
Cladophlebis ladinica	24	4.17						4.17	1		1			101	
Cladophlebis sp.	4														
Cone indet.	3														
Elatocladus sp.	1														
Equisetoid stem fragment	1														
Indet.	5														
Neuropteridium elegans	3														
Nilssonia cf. neuberi	40	10.00				10.00			1	1	4		4	12	
Nilssonia sp.	34	23.53	2.91			20.59	2.94		6	3	10	1	9	1;2;7;12; 13;128	
Pelourdea sp.	4	25.00				25.00			1	1	1		1	12	
Phlebopteris fiemmensis	6	33.33	16.67	16.67		16.67			2	2	2	1	1	2;80	
?Podozamites sp.	17	17.65						17.65	2		3			72;100	
Pterophyllum sp.	1														
Radicites sp.	1														
Schizoneura paradoxa	6														
Scytophyllum bergeri	55	54.55	7.27	3.64	1.82	49.09			8	4	37	4	34	3;5;12;13; 14;40;63;80	
Seed	2														
Sphenozamites sp.	37	13.51	2.70			13.51			3	2	5	1	5	2;8;12	
Stem indet.	6														
Taeniopteris sp.	8	25.00				25.00			2	1	2		2	12;14	
Thaumatopteris sp.	3														
Voltzia sp. 1	84	3.57	1.19	2.38			1.19		2	2	3	1		48;121	
Voltzia sp. 2	41	2.44		2.44					1	1	1			121	
Voltzia sp. indet.	170	2.94		2.35		0.59			2	2	5		1	12;121	
Wood	3														
Total	684	12.14	1.32	1.61	0.15	9.36	0.29	0.73	20	7	95	9	75	1;2;3;5;7; 8;12;13;14; 17;29;40;48; 63;72;80;100; 101;121;128	
Notes.

DMG percentage of damage

Spec Specialized damage

PS Piercing and sucking

Ovi Oviposition

FFG Functional Feeding groups

DTO Damage type occurrence

The most recent approach toward understanding the patterns of herbivory in the fossil record involves quantification of both the richness and intensity of insect damage (Wilf & Labandeira, 1999; Labandeira, Johnson & Lang, 2002; Labandeira et al., 2007; Kustatscher et al., 2014). The richness of herbivory is determined first by establishing a classification system of distinctive, diagnosable damage types, or DTs, that can be used generally in studies of herbivore damage to plants. DTs then are grouped into functional feeding groups (FFG). Eight functional feeding groups are present in the Monte Agnello flora ((i) external foliage feeding, subdivided into hole, margin, surface feeding and skeleotization; (ii) piercing and sucking; (iii) oviposition, though not truly a feeding interaction but rather egg-laying that leaves a significant record of plant damage; (iv) mining and (v) galling). To date, over 290 fossil DTs have been identified (CC Labandeira, pers. comm., 2014). Finally the DTs are ranked by their host specificity (HS), ranging from 1 for generalists to 3 for high host-plant specialization, which then allows non-generalized DTs (e.g., those with HS of 2 and 3) to be analyzed separately.

Each foliar element was photographed using a Canon EOS 30D camera with a Canon EF-S 60 mm f/2.8 macro lens or a Nikon Coolpix E4500. All photographs were optimized using Abobe Photoshop CS6 and Adobe Lightroom 5.

Quantitative analysis

Quantitative analyses of insect damage were done using R version 3.1.0 (www.r-project.org). For damage diversity analyses, sample size was standardized by selecting random subsets of foliar elements without replacement and calculating the damage diversity for the subsample. Subsets of the data were subjected to rarefaction using an analytic method detailed below, which extends the solution found by Wappler et al. (2012) to cases where individuals may belong to multiple classes and allows the explicit reconstruction of probability distributions for the rarefied sample (Heck, van Belle & Simberloff, 1975). This process was repeated 5,000 times, and the results were averaged to obtain the standardized damage diversity for the bulk flora and four single sub-localities (MA1, MA5, MA7, MA8). The remaining sub-localities were removed from the census because the target sample size of at least 40 specimens was not reached. The standard deviations (SD) for the resamples were calculated to provide sample error bars.

Results

Damage on the bulk Monte Agnello flora

Of the 684 plant remains examined from the Monte Agnello flora, 83, or 12.13%, exhibit some sort of damage represented by 20 different damage types. The taxa or morphotypes examined were represented by foliage, axes, stem fragments, fructifications, and dispersed seeds (Table 2). A total of 95 damage type occurrences were observed throughout the bulk flora: 45 on cycadophytes (representing 36.5% of all specimens), 37 on seed ferns (8.0%), ten on conifers (44.3%), and three on ferns (7.6%) (Table 3), suggesting that selective feeding by insect herbivores preferentially targeted particular seed plants. This pattern of selectivity was also recognized within the early late Permian (Wuchiapingian) of the Gröden/Val Gardena Sandstone from the Bletterbach Gorge of the Dolomites (Northern Italy) (T Wappler, pers. obs., 2013). Herbivory recorded for the Monte Agnello sites represents nearly all of the fundamental modes of herbivory, excluding fungal infection, which was not observed (see Gunkel & Wappler, 2015). Multiple DTs or functional feeding groups were only recorded in 1.6% of the plant remains whereas the majority were only damaged in one way (∼11%). Seven distinctive functional feeding groups have been detected on the foliar elements from Monte Agnello, most of which occur on particular plant hosts. Types of the external foliage feeding constitute 78.9% of all DT occurrences and preferentially occurred on the seed fern S. bergeri and consists of the exophytic consumption of live plant tissues, subdivided into skeletonization and margin-, hole- and surface feeding; this is the most common ensemble of Triassic damage types (Labandeira & Prevec, 2014; T Wappler, pers. obs., 2013) (Fig. 2). Those of the galling FFG provided 11.5% of all DT occurrences and are more or less evenly distributed among conifers, ferns and seed ferns (Figs. 3C–3D and 3G). Galling represents the most biologically complex of all major interactions, and represents arthropod-induced abnormal cell proliferation that can occur on all major plant organs (e.g., Kustatscher et al., 2014; Scott, Anderson & Anderson, 2004); examples are widely known (e.g., Stone & Schönrogge, 2003). Oviposition, though not a feeding interaction, comprised 5.2% of all DT occurrences; examples are common (Ghosh, Kar & Chatterjee, 2015; McLoughlin, 2011) (Figs. 3E–3F and 1). Minor levels of insect damage were present for piercing-and-sucking (2.1% of all DT occurrences; Fig. 3J) and mining (1.1%; Fig. 3H) FFGs. Leafminers construct distinct leaf mines, most of which are quite conspicuous and represent a form of endophagous herbivory in which a herbivore targets and feeds on fluid tissues such as phloem, mesophyll or epidermal cell protoplasts (Sinclair & Hughes, 2010); examples are uncommon and the possible mining structure on the pteridosperm Scytophyllum bergeri (Fig. 3H) indicates that the origin and diversification of the leaf-mining habit occurred about 92 million years before the first appearance of fossil angiosperms (Ash, 1997; Gnaedinger, Adami-Rodrigues & Gallego, 2014; Kustatscher et al., 2014; McLoughlin, 2011; Moisan et al., 2012; Pott et al., 2008).

Figure 2 Examples of external foliage feeding at Monte Agnello (Dolomites, N-Italy).

(A), Scytophyllum bergeri Bornemann, 1856 with intensively consumed leaf margins (DT12, 14) (MGP63/97). (B) Hole feeding indicated by leaf removal on both sides of the primary veins (DT63) on S. bergeri Bornemann, 1856 (MGP196/39A-B). (C)–(D) Hole feeding on a Sphenophyte (DT8) (MGP194/106), enlarged in (D) (E) Marginal feeding on the cycadophyte Nilssonia cf. neuberi Stur ex Pott, Kerp & Krings, 2007 (DT12) (MGP191/6A). (F) Excision of leaf to primary vein (DT14) on Bjuvia cf. dolomitica Wachtler & Van Konijnenburg-van Cittert, 2000 (MGP181/11A). (G) Removal or abrasion of surface tissues with a weak reaction rim (DT29) indicated by the dotted lines on B. cf. dolomitica Wachtler & Van Konijnenburg-van Cittert, 2000 (MGP196/43). (H) Cuspate excision (DT81) on S. bergeri Bornemann, 1856 (MGP171/28), enlarged in (I). (J)–(L), External foliage feeding on B. cf. dolomitica Wachtler & Van Konijnenburg-van Cittert, 2000 (MGP195/69A), deep excision of leaf margin enlarged in K (DT12) and interveinal tissue removed in L (DT17). Scale bars: striped, 10 mm; solid, 5 mm; dotted, 1 mm.

Figure 3 Examples of internal foliage consumption at Monte Agnello (Dolomites, N-Italy).

(A)–(B) Elliptical piercing and sucking punctures on the conifer Voltzia sp. 1 (MGP196/35), enlarged in (B) (DT48). (C) Ellipsoidal, sessile bud gall from branchlet (DT121) on the unaffiliated Voltzia sp. 1 (MGP171/81). (D) Small, hemispherical, thoroughly carbonized structures (DT80) on Phlebopteris fiemmensis Kustatscher et al., 2014 (MGP181/57C), indicated by arrows. (E) Fern Speirocarpus sp. (MGP197/69B) showing lenticular-ovoidal foliar oviopsition scars (DT101), indicated by arrows. (F) and (I) Lenticular-ovoidal foliar oviopsition scars (DT100) on the unaffiliated cycadophytes (MGP196/6; MGP196/7A). (G) Undifferentiated galling structures (DT80) on a seed-fern (MGP63/94), indicated by arrows. (H) Semilinear, frass-laden, mining structure with a smooth and rimmed margin (DT40) on Scytophyllum bergeri Bornemann, 1856 (MGP63/98A), asterisk indicates initial place of oviposition. (J) Ellipsoidal scale impressions with roughened surface (DT128) on the cycadophyte Nilssonia cf. neuberi Stur ex Pott, Kerp & Krings, 2007 (DT128) (MGP194/72A). Scale bars: striped, 10 mm; solid, 5 mm; dotted, 1 mm.

Table 3 Floral and insect damage composition of the late Ladinian flora from Monte Agnello, Dolomites, Italy on higher classification level.

Plant groups	# Leaves	% DMG	% Spec	% Gall	% Mine	% External	% PS	% Ovi	DTs	# FFGs	DTO all	DTO spec	DTO external	DT numbers	
Conifer	303	1.00	0.33	2.31		0.66	0.33		3	3	10	1	2	12;48;121	
Cycadophytes	250	16.00	1.20	0.40		13.60	0.40	1.60	14	6	45	3	38	1;2;3;7;8; 12;13;14;17;29; 72;80;100;128	
Indet.	16														
Ferns	52	5.77	1.92	1.92		1.92		1.92	3	2	3	1	1	2;80;101	
Seed ferns	55	54.55	7.27	3.64	1.82	49.09			8	4	37	4	34	3;5;12;13;14; 40;63;80	
Sphenophytes	8														
Total	684	12.13	1.32	1.61	0.15	9.36	0.29	0.73	20	7	95	9	75	1;2;3;5; 7;8;12;13; 14;17;29;40; 48;63;72; 80;100;101; 121;128	
Notes.

DMG percentage of damage

Spec Specialized damage

PS Piercing and sucking

Ovi Oviposition

FFG Functional Feeding groups

DTO Damage type occurrence

Damage on individual species

Among the 28 taxa represented at Monte Agnello less than half indicate some kind of damage, whereas, three—Scytophyllum bergeri, Bjuvia cf. dolomitica and Nilssonia sp.—are the most herbivorized taxa (71,6% of all DT occurrences) but only representing one-third of the flora (Table 2). The most abundant plant species are the conifers Voltzia sp. (Fig. 3C) and Voltzia sp. 2, which have the lowest damage frequency (2.44–2.94%) of the common Monte Agnello taxa. Ferns are nearly equally as diverse as the seed ferns but damage frequency is at least ten times less abundant than among the seed ferns (Table 3 and Fig. 3D). Sphenophytes displayed no signs of insect-mediated herbivory but the small number of sampled leaves open the possibility that more collecting and study may yet reveal damage to this group also.

Damage at distinct sub-localities

Plant material is generally preserved at the base of the “explosion breccia” at an angle to the bedding rather than compacted into a single horizon. Transport distance, therefore, must have been short, and burial was likely rapid. Thus, the fossil leaf assemblages must be considered as para-autochthonous (e.g., Hanley et al., 2007). Minimal transport allows us to document considerable changes in species composition and insect folivory over short distances and recognize possible heterogeneity in the structure and composition of the source plant communities and their associated herbivores. Large-scale disturbances may profoundly alter the composition and structure of plant communities and are rarely uniform in their influence on vegetation (Kustatscher, Dellantonio & Van Konijnenburg-van Cittert, 2014). Variations of floral composition and insect herbivore damage at the four sub-localities (MA1, MA5, MA7, MA8) censused are shown in Fig. 4 and Table 4. MA1 has the highest floral diversity (22 ssp.), followed by MA5 (16 ssp.) and MA7 (15 ssp.). MA8 is an extremely low-diversity flora (7 ssp.). Interestingly, all sites are strongly dominated by a single plant group representing in all cases over half of the characteristic plant material at that site. The most abundant plant lineage at MA5–MA8 is conifers, whereas at MA1 57% of the taphocoenosis is composed of cycadophytes. However, when analyzing herbivory on individual host groups at the four sub-localities, total damage frequency and external foliage feeding is overwhelmingly found on seed-fern hosts (Fig. 4), except MA7 where the preferred host-plants are cycadophytes.

Figure 4 Plant and damage composition within the single sub-localities.

Pie charts showing the frequency specimen data by (A). Host plant abundance (pooled in higher taxonomic ranks). (B)–(D) Damage composition. MA1, MA5, MA7, MA8, fossil sites.

Table 4 Floral diversity and evenness.

Flora	N	S	Rarefied species diversity at 40 leaves	Rarefied external damage diversity at 40 leaves	Rarefied specialized damage diversity at 40 leaves	Pielou‘s J	Simpson D	
MA1	244	22	12.89 ± 1.52	2.30 ± 1.13	0.63 ± 0.69	0.89	1.89	
MA5	236	16	9.03 ± 1.41	3.19 ± 1.08	0.68 ± 0.75	0.73	1.83	
MA7	125	15	9.38 ± 1.47	1.29 ± 0.49	na	0.63	1.70	
MA8	44	7	6.85 ± 0.36	0.93 ± 0.25	na	0.75	1.71	

Discussion

Volcanogenic deposits can preserve spatio-temporal biotic patterns at levels of resolution not commonly represented in the fossil record. Consequently, the plant–insect assemblages recognized in this study appear compositionally and ecologically unique (Currano et al., 2011; Dale, Swanson & Crisafulli, 2005). The para-autochthonous early late Ladinian flora of the Monte Agnello (Dolomites, N-Italy) offers insights into the patterns of arthropod herbivory during the beginning of the third pulse of herbivore expansion (sensu Labandeira, 2006a; Labandeira, 2006b; Labandeira & Currano, 2013: Fig. 1). It also provides insights into the way herbivores responded to environmental perturbation and the reorganization of community structure. Even though our data are preliminary, the palaeoecological and temporal setting of the early late Ladinian flora of the Monte Agnello in the Dolomites supports three major conclusions that parallel those drawn from data known from intensively studied Gondwanan sites.

(1) Dominance of seed plant herbivory. The dominance of seed plant herbivory by local arthropod herbivores, particularly that known since the Permian across western Euamerica (e.g., Schachat et al., 2014), Europe (Geyer & Kelber, 1987; T Wappler, pers. obs., 2013), Cathaysia (Glasspool et al., 2003), and in the extensive glossopterid-dominated floras across Gondwana (e.g., Cariglino & Gutiérrez, 2011; McLoughlin, 2011; Prevec et al., 2009) is also a conspicuous component of the late Anisian to Ladinian environments. This documents the persistence of the preferential targeting of selected groups of seed plants, like the cycadopytes in Monte Agnello, particularly by external foliage feeders. The pattern could be interpreted to support Feeny’s apparency hypothesis (Feeny, 1976), as seed plants were the most abundant and conspicuous, and therefore would have been the most apparent to herbivore consumption. However, for the Monte Agnello data, a more likely explanation favors increased herbivory on particular plants due to the anatomy of their leaves, suggesting that particular physical traits, like the scleromorphic structures of conifer taxa, reduce the palatability and digestibility of such plant material or act as a deterrent when more palatable plants are available (Labandeira & Anderson, 2005).

(2) Increase of interactional diversity and rise of the leaf-mining habit. There is an increase in plant–insect interactional diversity during the Early to Late Triassic in eastern Euamerica and Gondwana regions (e.g., Kustatscher et al., 2014; Scott, Anderson & Anderson, 2004), coupled with an increase in the diversity of FFGs, DTs, and associated herbivore behaviors observed at Monte Agnello, compared to insect damage from earlier known floras (e.g., Kustatscher et al., 2014). Of particular importance is the presence of the leaf-mining habit in which holometabolous insect larvae consume the inner parenchymal, epidermal, vascular, or other tissues of a plant, leaving the outer wall of the epidermis undamaged (Hering, 1951). The earliest documented leaf-mining fossil records have been reported from Kyrgyzstan, Austria, Australia and South Africa in deposits of Middle to Late Triassic age (comp. Table 1).

(3) Volcanic activity and site-specific habitat differences. The data presented here show that volcanogenic deposits are valuable for the creation and preservation of in situ sequential stages of biotic change not commonly represented in the fossil record. These episodic volcanic activities directly influenced the evolution of the environment, spatial structure and temporal dynamics of the plant community and the herbivores associated with the plants, resulting in vegetational heterogeneity had impact on both the likelihood and strength of interactions between plants and insect herbivores (e.g., Agrawal, Lau & Hambäck, 2006; Currano et al., 2011). Therefore, the heterogeneity among the sub-localities indicates that volcanic disturbance caused compositional and structural changes in the ecosystem during the time it occupied the site, which explain variations in plant physiognomy, plant and insect herbivore composition, and the overall paleoecology (Table 4). This conclusion is supported by (1) the spatial variability in the percentage of herbivorized plant host specimens, (2) the elevated number of DTs on each host plant, and (3) the differences in evenness and the relative abundance distributions of damage among the single sub-localities.

These conclusions warrant further verification from investigations of additional new sites to clarify patterns of arthropod herbivory during this crucial period of time where terrestrial ecosystems were beginning to become modern.

We thank Federica Angeli (Trento, Italy), Fulvio Boninsegna (Predazzo, Italy), Andrea Braito (Daiano, Italy), Daniele Ferrari (Museo Geologico delle Dolomiti, Predazzo, Italy), Christian Fontana (Vigo di Fassa, Italy) and Guido Roghi (CNR and University of Padova, Italy) for their help during fieldwork and the preparation of the fossils. Special thanks goes to Daniele Ferrari for the photographs of the plants. The authors acknowledge Ellen Currano, University of Wyoming, an anonymous reviewer, and the editor William DiMichele for their constructive and encouraging comments.

Institutional abbreviations

MGP Museo Geologico delle Dolomiti Predazzo Specimens occurring on the same rock slab are identified by different capital letters following the catalogue number.

Additional Information and Declarations

Competing Interests

Author Contributions

Elio Dellantonio works for the Museo Ceologicao dell Dolomiti, and Evelyn Kustatscher is employed at the Naturmuseum Südtirol.

Torsten Wappler conceived and designed the experiments, performed the experiments, analyzed the data, contributed reagents/materials/analysis tools, wrote the paper, prepared figures and/or tables, reviewed drafts of the paper.

Evelyn Kustatscher contributed reagents/materials/analysis tools, wrote the paper, prepared figures and/or tables, reviewed drafts of the paper.

Elio Dellantonio contributed reagents/materials/analysis tools, prepared figures and/or tables, reviewed drafts of the paper.

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
