# Peer review of "Plant–insect interactions from Middle Triassic (late Ladinian) of Monte Agnello (Dolomites, N-Italy)—initial pattern and response to abiotic environmental perturbations"

_PeerJ, doi:10.7717/peerj.921_

## Round 0.1 · original submission · Major Revisions

· Academic Editor

Major Revisions

I believe this to be a significant paper; consequently, I have gone through the paper in some detail and made suggestions on both the English and the content, independently of the comments made by the reviewers. These are in a separate PDF file of the manuscript that will be sent from PeerJ editorial staff.

Although the reviewers call for "major revision", which which I essentially agree, the paper nonetheless appears to be in very good shape and will bring an important, and expanding area of paleontological research to PeerJ. Thus, by "major revision" I would suggest that the attention the paper needs is more than touch-up, and that there are some substantive matters to consider. But not that there are great flaws in the logic or presentation.

>A couple of points with regard to the reviews.

-The English does need tightening, and I have made such suggestions as I could (in comment bubbles, referring to text highlighted in yellow).

-I believe Reviewer 2 misunderstood the references to colonization of the land, which does indeed, as I interpret it, refer to the initial colonization, not the post P-Tr reoccupation of the land surfaces that might have been devastated by environmental perturbation. I have suggested some rewording to make this clear.

-It is not good to cite papers that are in preparation. These could be cited as personal communications if vital to the paper.

-There is a problem, it seems to me, with pinning the importance of this paper on it being an example of the recovery from the P-Tr events. It is a long time after that event. Also, to the extent I understand it (not being an expert on the Triassic) we actually know very little about floras of the Early and Middle Triassic. Although not unknown, there really is no "trajectory" that can be referred to with confidence.

-There also is a problem with the term "recovery" - this implies a return to an equilibrium, or at least a measurable point. In fact, Triassic floras are different from those of the late Paleozoic. Thus, it is not really a "recovery" but a more complex matter of revegetation of (presumed) devastated landscapes from (presumed) refugial areas...or do I have this wrong? If I do, then the authors might consider saying more about this sort of thing so that those with some familiarity can interpret the goings-on more accurately.

-I cannot comment on the matter of "delayed recovery" and the authors certainly should not be held to the comments of the reviewer. They should, however, if they choose not to follow this reviewer's suggestion, explain why, and perhaps consider addressing this in the paper.

-The authors should, of course, address the comments of the reviewers if they choose to revise the paper and resubmit it. Even if they find they disagree with reviewer comments, they offer examples of the kinds of objections readers may pose and thus can be preemptively addressed.

·

Basic reporting

This manuscript is “self-contained,” an “appropriate unit of publication,” and to the best of my knowledge adheres to all PeerJ policies. The overall structure is fine, although see specific comments below and in the following sections. Figures are well done, and the photographs in Figures 2 & 3 are especially nice! I have selected "major revisions" because while the data are well-collected and analyzed, the framing of their context and interpretations of results needs improvement.

I have four main suggestions for improvement in the basic reporting category:

1) Rework the introduction to more clearly state the context and significance of the study. The Monte Agnello flora provides insight on four main topics: Labandeira’s proposed pulses of herbivore expansion, Triassic floral change in Europe (lines 217-225 are better suited for the introduction than the discussion), Permo-Triassic biotic crisis, and investigation of local perturbations and heterogeneity across a landscape. Each deserves its own full paragraph in the introduction, with a little more detail on each point. For example, for #1, briefly summarize what the four phases of herbivore expansion are and provide a little more detail what is already known from other parts of the world about the 3rd phase.
2) Similarly, the results section can also be better structured. Have separate paragraphs that report 1) damage on the bulk Monte Agnello flora, 2) damage on individual species, and 3) damage at distinct sites. Additionally, make sure to keep the results and the interpretations in separate sections. Lines 182-184 (beginning with “likely a consequence”) and 202-208 are interpretations of the results and belong in the discussion.
3) Figure 4 needs a caption that clearly describes what is represented in each column of pie charts. For example, does the first column of pie charts refer to percent of specimens or species? The label “plant diversity” implies it is species, but specimens would be more informative in comparison with the other 3 columns.
4) As a scientific reviewer, my primary objective was to analyze the science presented and the overall structure of how it is presented. However, I note that the writing (grammar, word choice, clarity of sentences) needs improvement prior to final publication.

Experimental design

This submission represents original research conducted according to high technical standards, and the methods are sufficiently described to allow for reproducibility.

Validity of the findings

The introduction presents the Monte Agnello flora in the context of the four main themes, but these are not adequately reflected upon in the discussion. If these themes are to be retained, I’d like more explicit discussion of how the new data fits in. Below are comments on each theme:
a. Pulses of herbivore expansion: Give more specifics on how Monte Agnello relates to Paleozoic and other Triassic sites. How do these data (number of DTs, specificity, endophagy vs. exophagy, targeting of seed ferns, scarcity of damage on conifers) compare with late Paleozoic sites? With other Triassic sites?
b. Triassic floral change in Europe: This is well discussed at various points in the text. Only comment is to improve the organization such that all comments on this are together.
c. Permo-Triassic Biotic Crisis: Care must be taken in using Monte Agnello to evaluate the role of the P-Tr perturbation on European ecosystems. Neither Late Permian (baseline for pre-extinction conditions) nor Early Triassic (immediate response of biota) data are presented for comparison. In particular, tone down the language in lines 59-61.
d. Volcanic activity and site-specific habitat differences: More information is needed to use Monte Agnello to illustrate the effects of volcanic activity and site-specific habitat differences. The sites are preserved in tuffs at the base of the explosion breccia, but are they stratigraphically equivalent? What can be said about the depositional environment of each site? Are there any geologic differences among the sites that could provide insight on differences in floras or herbivory among sites? Also useful would be estimates of how much time and space are represented by each flora. Are these in situ landscapes being instantaneously buried by the air-fall tuff, or was hot, unconsolidated ash traveling down topography, mixing and transporting plant material? What is the evidence?

Additional comments

I have line by line comments from my reading of the manuscript:

There are several references to manuscripts in preparation (e.g. lines 25, 153-155, 164). I do not know whether PeerJ allows this; many journals do not. Personally, I advise against them because they are not available to the reader to make comparisons for him/herself.

Lines 29-33: Awkward sentence structure. It sounds as though the Paleozoic, rather than the P-Tr boundary marks the start of the third pulse of herbivore expansion.

Lines 47-55: It is unclear why Monte Agnello is so distinct from the other floras. It is also dominated by conifers and has few ferns, seed ferns, horsetails, or lycophytes. Is it the high representation of cycadophytes?

Line 53: Recommend revising to “most diverse late Middle Triassic biota in the region.” There are not enough data presented to compare Monte Agnello to the Molteno, for instance.

Line 54: Use of the word “elevated” – is 9.36% high? It would be helpful to have more hard numbers comparing this to other sites. Perhaps in the Table 1, you could add numerical data (% of leaves damaged, % area damaged, etc) to help put the Monte Agnello data in context of other Triassic studies.

Line 68: What is meant by “only influenced”? Do you mean to say that the rocks at Monte Agnello are all the result of the Predazzo volcano? Please clarify.

Line 102: Please add numerical area covered by the sites.

Line 109-110: Lower case and upper case rather than minuscule and mainuscule.

Line 112: Qualitative analysis . I think this can simply be part of the previous section and does not need its own heading.

Line 113-115: Since the Wilf/Labandeira work came first, it is odd to call these an extension of Kustatscher et al. Instead of “and extensions by”, perhaps “building upon.”

Line 144-145: Unclear sentence.

Line 151-152: In addition to giving the number of DTOs on each major plant group, it would also be interesting to see right there what percent of the flora each plant group represents. That would make the strong case for selective feeding, especially if insects are targeting the non-dominant plant groups!

Line 164: Include a reference to the Figure 3 images of galling.

Line 170-177: More clearly separate the discussion of piercing and sucking and mining. As written, it is difficult for a non-expert to distinguish the two, and mining sounds like an exophytic interaction.

Line 178-180: While the three taxa that have 71.6% of the DTOs together make up just a third of the total flora, they are the #2, #4, and #7 (? Might have counted that last one wrong) most abundant species. Given Feeny’s ideas on apparency, it makes perfect sense those would have lots of herbivore damage.

Line 186: I wouldn’t read much into no damage observed on sphenophytes. There are only 8 specimens, and with a ~12% overall damage frequency, it would not be surprising to have no damaged leaves.

Line 195-196: Give also the rarefied plant species diversity, since these allow better comparisons among sites of vastly different sample sizes.

Tables 2 & 3: Not necessary to report Indet. specimens. (These weren’t used in analyses, were they?)

Reviewer 2 ·

Basic reporting

This is a generally well written account of insect damage in Middle Triassic (late Ladinian) fossil plants of the Italian Dolomites, and is one of the few studies to give comprehensive statistics on proportions of different damage types . Many comparable studies seem to document only one kind of damage, so this paper seems of unusual interest.

I am not a fan of acronyms such as DT, HS and FFG which do not appear that often anyway.

Papers in preparation or submitted should not be cited (l.164). In press citations should only be allowed when posted online.

What is ladder in l. 224?

Experimental design

The authors take pains to find a connection to the Permian-Triassic life crisis, but I find this rather strained because the Ladinian is more than 10 million years after that event. It is also about 7 million years after the Spathian extinctions, which was also a big one associated with a post-apocalyptic greenhouse event. My preference would have been to see some relationship with sedimentary disturbance: relating the damage to ecological successiobnal status and disturbance level as determined indpendently by analysis of associated tuffs, sedimentary structures and paleosols. If the Permian-Triassic is the real problem, then some time series of damage changes from Permian to Middle Triassic would be needed, and I am not sure adequate data is available for that.

Validity of the findings

The damage types appear well founded and the proportions and their variance reasonable: the wider implications of the work for the Permian-Triassic transition are not well supported.

The term "colonization of land" is not appropriate to the Permian-Triassic (l.20), more for Silurian.

The idea of "delayed recovery" (l. 43) has now been debunked in numerous separate papers by Payne, Bottjer and Retallack, who demonstrate multiple hits in Changsingian, Griesbachain, Dienerian, Smithain and Spathian.

Ectophytic and endophytic are new terms for me (l.56), but I wonder if they are needed since both terms are used that would seem to cover everything.

Is Chiropteris a fern? (l. 90). The comparable Rochipteris is a gymnosperm (Kannaskoppiaceae)

---

## Round 0.2 · Minor Revisions

· Academic Editor

Minor Revisions

I believe this paper is acceptable with minor editorial revisions, and possibly some textual alterations, as suggested by the reviewer. I do not believe it requires any further review.

The editorial comments I have offered are entirely with regard to English language expression, which, overall, is quite good throughout most of the manuscript. There are spots, however, where clarity is called for. Of course, my intent was not to change meaning, which I am sure the authors will monitor carefully.

The authors may deal with the reviewer comments as they see fit. These comments clearly are offered in a helpful spirit, and they raise some interesting points. The high diversity in the conifer community may be a bit more of a botanical point than is required to be addressed in this paper, but I would leave that to the discretion of the authors.
To clarify, in case there is any confusion, I only requested a single review, from one of the previous reviewers, to be certain that my opinions of the scientific soundness of the paper were correct.

·

Basic reporting

Since this is a re-review, I am putting all of my comments here rather than dividing them. In the "comments to author" box, I have included some of my additional thoughts on the data and discussion, but addressing these are not necessary for publication.

In short, I am delighted to see how much this manuscript has improved since the initial submission and think it is ready for publication in PeerJ once a final check has been done on the grammar. The revised introduction and discussion do a much better job of framing the importance of the Monte Agnello flora, with the emphasis on phases of herbivore expansion and impact of volcanism across a landscape and only a brief mention of the PT extinction. I am also very happy with the expanded discussion of the geology and depositional settings of the floras.

A few minor comments, which should be fixed before publication:

Line 135: Should be either 8 or 5 functional feeding groups (not 7), depending on whether external foliage feeding is counted as one thing or the four subdivisions are each being counted separately

Line 161-162: This sentence is unclear. How is this different from what is explained in the rest of the paragraph? Did you mean to say that you also resampled by site? By taxon?

Line 174: Eliminate the word, "thus." There's not a clear and strong tie to the previous sentence.

Line 190: Report percents for mining and piercing and sucking separately. This could simply be done by saying: "Minor levels of insect damage were present for the piercing-and-sucking (X.X% of all DT occurrences; Fig. 3J) and mining (Y.Y%, Fig. 3H) FFGs.

Line 247: eliminate seed ferns, since they represent a small % of the total flora and do not make a strong case for apparency, unlike the cycadophytes.

Figure 4: The caption is improved, but I think it could still be clearer by explicitly spelling out each panel. And, rather than labeling A “plant diversity,” “plant composition” might be more clear since it’s the percent of specimens belonging to each taxonomic group, not the number of species.

Experimental design

No comments

Validity of the findings

No comments

Additional comments

The comments below are not critiques of the paper, just three things that occurred to me while reading.

1) It is interesting to me that all sites are strongly dominated (>50%) by a single plant taxon. This is something I've observed in nearly all the angiosperm sites that I've collected, and I've always wondered whether we are actually seeing the true forest, whether we are really biased towards the couple trees that grew immediately above the quarry site, or whether there is some sort of preservational bias. Just thought I'd share, in case you have thoughts on this.

2) MA1 is the most diverse site, and also the only one where conifers are not very abundant. Might there be a connection?

3) Also, are there any interpretations you can make about what parts of the landscape the different assemblages represent? What environments or successional stages were dominated by cycadophytes vs. conifers? This may not be known, but if you can say anything about it, it would be very interesting to do so in the manuscript.

---

## Round 0.3 · accepted · Accept

· Academic Editor

Accept

The authors are to be thanked for the rapid turnaround and careful revision of the manuscript. It is now, I believe, ready to move forward to the production stage.